# Efficient Multi-modal Long Context Learning for Training-free Adaptation

**Zehong Ma** [1]   **Shiliang Zhang** [1 2 *]   **Longhui Wei** [3]   **Qi Tian** [3 4]

## Abstract

Traditional approaches to adapting multi-modal large language models (MLLMs) to new tasks have relied heavily on fine-tuning. This paper introduces Efficient Multi-Modal Long Context Learning (EMLoC), a novel training-free alternative that embeds demonstration examples directly into the model input. EMLoC offers a more efficient, flexible, and scalable solution for task adaptation. Because extremely lengthy inputs introduce prohibitive computational and memory overhead, EMLoC contributes a chunk-wise compression mechanism combined with layer-wise adaptive pruning. It condenses long-context multimodal inputs into compact, task-specific memory representations. By adaptively pruning tokens at each layer under a Jensen-Shannon divergence constraint, our method achieves a dramatic reduction in inference complexity without sacrificing performance. This approach is the first to seamlessly integrate compression and pruning techniques for multi-modal long-context learning, offering a scalable and efficient solution for real-world applications. Extensive experiments on diverse vision-language benchmarks demonstrate that EMLoC achieves performance on par with or superior to naive long-context approaches. Our results highlight the potential of EMLoC as a groundbreaking framework for efficient and flexible adaptation of multi-modal models in resource-constrained environments. Codes are publicly available at https://github.com/Zehong-Ma/EMLoC.

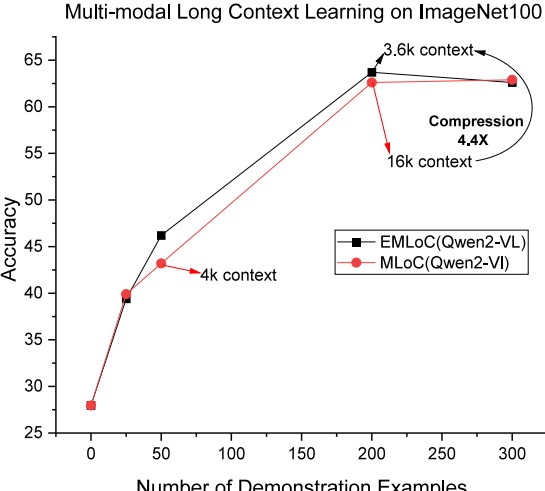

*Figure 1.* The comparison between EMLoC and MLoC on ImageNet100 with varying numbers of demonstration examples. With 200 examples, EMLoC achieves 4.4× context compression over vanilla MLoC without performance loss. It significantly outperforms MLoC with 50 examples using a similar context length.

## 1. Introduction

In recent years, multi-modal large models (MLLMs) (Liu et al., 2024; Zhu et al., 2024; Bai et al., 2023) have achieved significant advancements, demonstrating remarkable success across a wide range of multi-modal tasks. Traditionally, transferring these models to downstream tasks relies on supervised fine-tuning, including full fine-tuning and parameter-efficient methods (Hu et al., 2022). However, these techniques require updating model parameters and often incur substantial computational costs. Recently, MLLMs have evolved from processing single-image inputs (Liu et al., 2024) to handling multi-image and video data (Wang et al., 2024b; Hu et al., 2024; Chen et al., 2024c), while also supporting increasingly long context lengths.

A key observation, as illustrated in Figure 1, is that providing task-specific demonstration examples during inference significantly enhances model performance. We refer to this approach as Multi-modal Long Context learning (MLoC). However, directly feeding multiple multi-modal examples into the model often results in excessively long contexts,

---

[1]State Key Laboratory of Multimedia Information Processing, School of Computer Science, Peking University [2]Peng Cheng Laboratory, Shenzhen, China [3]Huawei Inc. [4]Guangdong Laboratory of Artificial Intelligence and Digital Economy (SZ). Correspondence to: Shiliang Zhang <slzhang.jdl@pku.edu.cn>.

*Proceedings of the 42nd International Conference on Machine Learning*, Vancouver, Canada. PMLR 267, 2025. Copyright 2025 by the author(s).

leading to prohibitively high inference costs.

To address these challenges, we introduce **E**fficient **M**ulti-modal **Lo**ng **C**ontext Learning for training-free adaptation (EMLoC), a training-free adaptation method. EMLoC leverages the benefits of multi-modal long context learning while mitigating its computational drawbacks through a chunk-wise compression mechanism combined with layer-wise adaptive pruning. This approach compresses extensive multi-modal contexts into compact, task-specific memory representations, significantly reducing computational overhead without compromising task performance.

The chunk-wise compression mechanism partitions a long context into smaller, manageable chunks, enabling a divide-and-conquer approach. This strategy reduces reliance on large-memory devices and significantly lowers computational overhead. Within each chunk, we employ a layer-wise adaptive pruning strategy, progressing from the top layer to the bottom layer. Using a greedy search algorithm, we determine the minimum number of tokens required per layer to ensure the Jensen–Shannon (JS) divergence between the original and pruned outputs remains below a predefined threshold. This approach avoids noticeable performance degradation while maximizing token reduction. Unlike static methods (Xiao et al., 2024; Li et al., 2024; Cai et al., 2024), which reserve a fixed number of tokens per layer, our adaptive strategy dynamically prunes tokens based on layer-specific importance, ensuring optimal efficiency without sacrificing accuracy.

The EMLoC framework operates without modifying model weights and requires only a few forward passes to generate task-specific compressed memory. This makes it a lightweight, plug-and-play solution that maintains task performance while drastically reducing inference costs. The contributions of this work are threefold:

- Efficient Multi-modal Long Context Learning: We introduce chunk-wise compression and layer-wise adaptive pruning to reduce computational costs and memory usage in long-context scenarios.

- Adaptive Layer Importance Analysis: We dynamically analyze the importance of different layers in a layer-wise manner, offering new insights into token pruning strategies.

- We establish a linear upper bound for the information loss and demonstrate the effectiveness of our method across diverse tasks with extensive experiments.

## 2. Related Work

This work is related to multi-modal large language models, long context learning, and training-free context compression.

We briefly review recent advances in these areas and discuss our contributions and differences with them.

### 2.1. Multi-modal Large Language Models

The progress in LLMs has propelled the advancement of MLLMs. Flamingo(Alayrac et al., 2022) pioneered the integration of a pre-trained visual encoder with the Chinchilla 70B(Hoffmann et al., 2022) LLM, demonstrating strong zero-shot and few-shot performance on vision-language tasks. Since then, numerous open-source models have emerged, including Kosmos-1(Huang et al., 2024b), MiniGPT-4(Zhu et al., 2023), and LLaVA (Liu et al., 2024). Subsequent research has expanded MLLMs' functional capabilities and improved their visual perception. Models like Kosmos-2(Peng et al., 2023), CogVLM(Wang et al., 2023), Shikra(Chen et al., 2023), Pink(Xuan et al., 2024), and LocLLM(Wang et al., 2024a) incorporate localization through the pix2seq paradigm or connections with detection and segmentation models. Others, such as Qwen-VL(Bai et al., 2023), Yi-VL(Young et al., 2024), DeepSeek-VL(Lu et al., 2024), InternVL(Chen et al., 2024c), and Intern-XComposer(Dong et al., 2024), enhance capabilities with high-resolution inputs and larger training datasets. Recently, models such as Intern-XComposer-2.5 (Zhang et al., 2024a), InternVL-2 (Chen et al., 2024c), MiniCPM-V-2.6 (Hu et al., 2024), and Qwen2-VL (Wang et al., 2024b) have advanced to support multi-image understanding, video comprehension, and multi-modal in-context learning. These recent studies have made it possible to explore MLLMs' multi-modal long-context learning.

### 2.2. Long Context Learning

In-context learning (ICL) is a good adaptation method to improve the performance of downstream tasks. We focus on multi-modal long context learning for training-free adaptation with many in-context examples.

**Scaling ICL.** The foundational study by (Brown et al., 2020) demonstrates that increasing the number of in-context examples improves the performance of LLMs. More recent works begin to explore the effects of scaling in-context examples beyond 1000, with (Li et al., 2023), (Agarwal et al., 2024), and (Bertsch et al., 2024) showing consistent performance improvements in multiple text-based tasks. However, these studies are limited to text-only benchmarks and lack exploration in multi-modal tasks.

**Multi-modal ICL.** Despite significant advancements in MLLMs, research on multi-modal in-context learning (ICL) remains in its early stages. Flamingo (Alayrac et al., 2022) pioneered the exploration of in-context capabilities in MLLMs, demonstrating few-shot learning across various vision-language tasks. Recent studies have evaluated the generalization abilities of models like GPT-4V and Gemini,

highlighting that in-context learning enhances performance on out-of-domain and out-of-distribution tasks (Zhang et al., 2024b; Han et al., 2023b). However, these works have not fully explored the potential of leveraging extended context windows to incorporate more demonstration examples and long-context inputs. A recent study by (Jiang et al., 2024) investigated many-shot in-context learning in MLLMs, revealing that open-source MLLMs struggle with complex long contexts, while closed-source models perform significantly better. This paper focuses on addressing the challenges of multi-modal long-context learning in open-source MLLMs, presenting an initial exploration of this domain. Our work aims to bridge the gap by enabling efficient and scalable long-context adaptation for open-source models.

### 2.3. Training-free Context Compression

Training-free context compression techniques have advanced significantly to address the challenges of handling extended sequences in LLMs. StreamingLLM (Xiao et al., 2024) retains Key-Value (KV) pairs from initial and recent tokens, leveraging attention concentration to reduce memory usage. SnapKV (Li et al., 2024) dynamically identifies critical KV pairs based on attention patterns, minimizing cache size while preserving performance. H2O (Zhang et al., 2023) prioritizes recent and high-attention tokens with a dynamic cache strategy, optimizing memory efficiency. PyramidKV (Cai et al., 2024) adjusts KV cache size hierarchically, allocating more memory to lower layers for efficient compression. Recent studies, such as LLaVolta (Chen et al., 2024a) and FastV (Chen et al., 2024b), have focused on compressing visual contexts to enhance the inference accuracy of multi-modal large language models. Despite these advancements in text-only and image-only context compression, multi-modal context compression remains relatively underexplored. In this work, we aim to adaptively compress multi-modal long contexts into an efficient compact memory without sacrificing performance.

## 3. Method

This section introduces our EMLoC for training-free adaptation. We begin with an overview of the approach, followed by the introduction of chunk-wise compression and layer-wise adaptive pruning. Finally, we present a theoretical analysis that establishes a linear upper bound on the information loss introduced by our compression strategy.

### 3.1. Overview

Given a downstream multi-modal task and its demonstration examples, our goal is to adapt a pre-trained MLLM without any training or parameter fine-tuning. To achieve this, we construct a long context $\mathbb{C}$ as the input of MLLM by concatenating $N$ task-specific demonstration examples $D$.

Representing each example as $\langle I_i, Q_i, A_i \rangle$ consisting of an image $I_i$, a question $Q_i$ and a corresponding answer $A_i$, the generation of $\mathbb{C}$ can be denoted as,

$$\mathbb{C} = \bigoplus_{i=1}^{N} \langle I_i, Q_i, A_i \rangle. \qquad (1)$$

Conditioning on the multi-modal long context $\mathbb{C}$ and multi-modal test queries $X$, MLLM can generate more accurate answers $Y$ as follows:

$$Y = \text{MLLM}(\mathbb{C}, X). \qquad (2)$$

Increasing the number of examples $N$ provides more informative contexts and enhances performance, but significantly increases the inference cost and memory usage.

We thus propose to reduce the long context $\mathbb{C}$ into a compact task-specific memory $M$ through chunk-wise compression with layer-wise adaptive pruning. Using $\mathbb{M}$ to denote the key-value (KV) cache of context $\mathbb{C}$, we aim to extract a more compact memory $M$ from $\mathbb{M}$.

This can be achieved by dividing the $\mathbb{C}$ into multiple chunks and adaptively spotting and reserving important tokens across chunks and layers to minimize the change of output probabilities distribution. The target of $M$ extraction from a chunk can be conceptually formulated as

$$M = \arg \min_{M \subset \mathbb{M}} |M|,$$
$$\text{s.t.} \quad \mathcal{D}_{\text{JS}}(P(Y \mid \mathbb{M}, X), P(Y \mid M, X)) \le \Delta, \qquad (3)$$

where $\mathcal{D}_{\text{JS}}(\cdot)$ denotes the Jensen-Shannon divergence, which measures the difference between two probability distributions. $\Delta$ is a configurable upper bound regulating the trade-off between information loss and the length of $M$.

The overall framework of EMLoC is illustrated in Figure 2(a). EMLoC implements the objective in Eq. (3) in two stages. It first partitions $\mathbb{C}$ into smaller chunks to enable processing on resource-constrained devices. As shown in Figure 2(b), we partition $N$ demonstration examples into $K$ chunks, each containing $\frac{N}{K}$ examples. We denote each chunk as $\mathbb{C}_k$, i.e.,

$$\{\mathbb{C}_k\}_{k=1:K} = \mathbb{C}. \qquad (4)$$

Let $D_k$ denote the individual demonstrations in chunk $\mathbb{C}_k$ before concatenation. We iteratively compress each chunk to form the compressed memory:

$$M_k = \text{LAP}(M_{k-1}, \mathbb{C}_k, D_k), \qquad (5)$$

where $\text{LAP}(\cdot)$ is our layer-wise adaptive pruning, and $M_{k-1}$ is the compressed memory from the previous chunks. Iterating this process from $k = 1$ to $k = K$ yields a final compressed memory $M_K$, i.e., $M$.

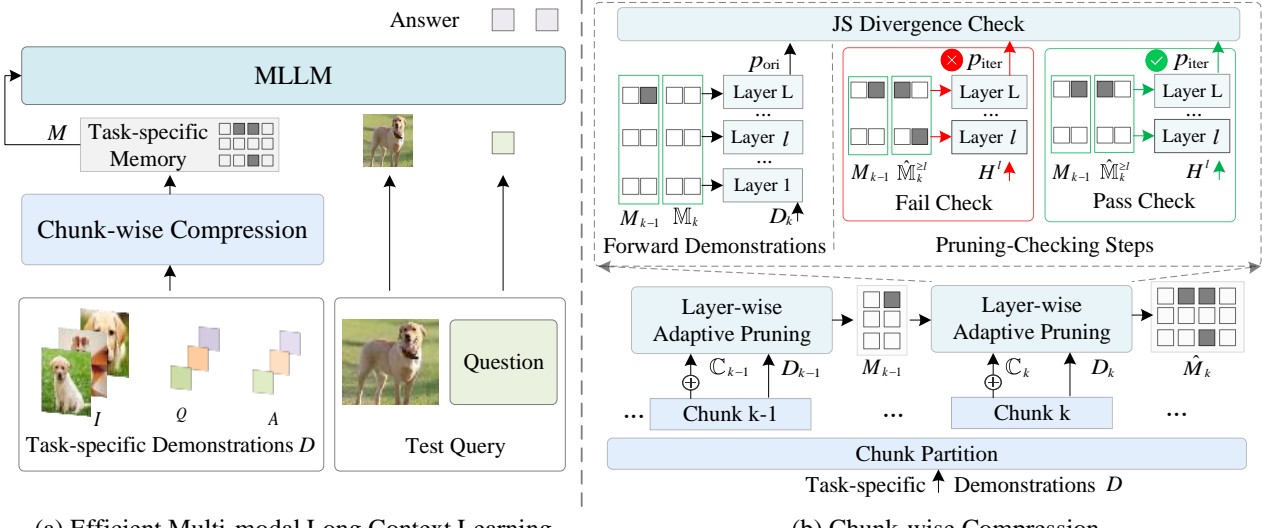

(a) Efficient Multi-modal Long Context Learning      (b) Chunk-wise Compression

*Figure 2.* (a) The overall framewrk of efficient multi-modal long-context learning. (b) Chunk-wise compression with layer-adaptive pruning, where pruning steps iteratively update output probabilities and are validated using a JS divergence check. Gray squares indicate pruned tokens, with red and green arrows representing failed and successful pruning steps, respectively.

EMLoC proceeds to prune the KV cache of each chunk in a layer-wise manner under a Jensen–Shannon (JS) divergence constraint, retaining only the most relevant tokens for the downstream task. It is difficult to set a global upper bound $\Delta$. An effective implementation to $\Delta$ is achieved by setting a local constraint $\delta$. The details of layer-wise compression and implementation to $\Delta$ are presented in following parts.

### 3.2. Layer-wise Adaptive Pruning

Different tasks require varying levels of knowledge, resulting in different degrees of token redundancy. Recent studies (Xiao et al., 2024; Cai et al., 2024) show that earlier layers in large language models are more crucial than later ones. Instead of a fixed pruning strategy (Li et al., 2024; Cai et al., 2024) that retains a constant number of tokens per layer, we propose a layer-wise adaptive pruning strategy that dynamically adjusts the token number per layer while preserving performance. The optimal number of tokens per layer is the minimum required to keep the Jensen-Shannon divergence between the probability distributions conditioned on the original and compressed memory below a predefined threshold $\delta$. Detailed pseudocode is provided in Appendix A. The procedure proceeds as follows:

**Forward Demonstrations:** Given the context $\mathbb{C}_k$ of $k$-th chunk, the KV cache $\mathbb{M}_k$ of this chunk is firstly extracted, conditioned on the previously compressed memory $M_{k-1}$:

$$\mathbb{M}_k = \text{ExtractKV}(M_{k-1}, \mathbb{C}_k). \tag{6}$$

A forward pass is performed through the MLLM with the concatenated memory $M_{k-1} \oplus \mathbb{M}_k$ and the individual demonstration examples $D_k \in \mathbb{R}^{\frac{N}{K} \times T}$ in chunk $\mathbb{C}$. The output includes the original output probabilities $p_{\text{ori}}$ of answer tokens, attention weights $\alpha \in \mathbb{R}^{L \times (\frac{N}{K} \times T) \times S}$, and hidden states $H \in \mathbb{R}^{L \times \frac{N}{K} \times T}$:

$$(p_{\text{ori}}, \alpha, H) = \text{MLLM}(M_{k-1} \oplus \mathbb{M}_k, D_k), \tag{7}$$

where $L$ is the number of layers, $T$ denotes the length of each demonstration example, and $S$ represents the sequence length of the current chunk. Then, for each layer, we calculate the importance score of each token in $\mathbb{M}_k$ using the accumulated attention weights from answer tokens. The answer tokens serve as an observation window to estimate the importance of each token in chunk $\mathbb{C}$. The importance score of $j$-th token in $l$-th layer can be formulated as:

$$\beta_j^l = \sum_{i \in \text{ans\_index}} \alpha_{ij}^l, \tag{8}$$

where ans_index is the index of answer tokens.

**Pruning-Checking Step:** Tokens with higher importance are retained, while less important ones are pruned iteratively from the top layer down. This top-down approach improves efficiency by avoiding forward passes through unpruned layers. At each step, pruning is applied to a single layer using a candidate ratio $r$, greedily selected in ascending order from the retention ratio set $R$. Once the top layers above the $l$-th layer are pruned, the pruning of the $l$-th layer is formulated as selecting the top $r \times S$ tokens based on importance scores:

$$\hat{\mathbb{M}}_k^l = \mathbb{M}_k^l(\text{Topk}(\beta^l, r \times S)). \tag{9}$$

As illustrated in Figure 2 (b), we can get the output probabilities $p_{\text{iter}}$ of all answer tokens in one pruning step:

$$p_{\text{iter}} = \text{MLLM}(M_{k-1}^{\geq l} \oplus \hat{\mathbb{M}}_k^{\geq l}, H^l), \qquad (10)$$

where $M_{k-1}^{\geq l}$ denotes the compressed memory from layer $l$ onward for the previous $k$-1 chunks, $\hat{\mathbb{M}}_k^{\geq l}$ denotes the compressed memory from layer $l$ onward for the current $k$-th chunk, hidden states $H^l$ is the input embeddings of $l$-th layer. Leveraging hidden states $H^l$ extracted in Equation (7) can reduce redundant computation in pruning steps.

After each pruning step, we perform the JS divergence check between the output distributions $p_{\text{ori}}$ and $p_{\text{iter}}$ to measure the information loss. If the divergence is below the threshold $\delta$, the pruning process for the current layer is successful and continues to prune the next bottom layer otherwise select a larger $r$ from rentention ratio set $R$ to prune this layer again. As shown in Figure 2(b), a step marked by red arrows fails the check, while a subsequent step with green arrows retains more tokens and successfully passes the check. This pruning-checking step ensures that the compressed memory retains the essential information for task performance while minimizing inference costs.

After successfully pruning all layer for $k$-th chunk, we can get the compressed memory $M_k$ of $k$ chunks by concatenating $M_{k-1}$ and $\hat{\mathbb{M}}_k$.

### 3.3. Theoretical Analysis of Information Loss

It is difficult to set a global upper bound $\Delta$ for JS divergence when sequentially processing multiple chunks. An effective implementation to $\Delta$ is achieved by setting a local constraint $\delta$ at each pruning-checking step. In this part, we theoretically analyze how the local constraint $\delta$ contributes to the global JS divergence $\Delta$ between the full memory $\mathbb{M}$ and its compressed counterpart $M$.

We assume that the $N$ demonstration examples $D$ follow the same data distribution as the test set. Thus, during inference with the memory $M$, the probability distribution $P_M$ of test set equals to that of demonstration examples $D$:

$$P_M = P(Y|M, X) = P(\pi_A(D)|M, \pi_{I,Q}(D)) = P_M^D, \qquad (11)$$

where $\pi_{I,Q}$ denotes extracting the images and questions as multi-modal questions and $\pi_A$ denotes getting the corresponding answers in the demonstration set.

**Local JS Distance Constraint:** With the individual demonstrations $D_k$ of the $k$-th chunk, the JS distance between the probability distributions under $M_{k-1} \oplus \mathbb{M}_k$ and $M_k$ is bounded by a divergence distance threshold $\sqrt{\delta}$:

$$\mathcal{D}_{\text{JS}}\left(P_{M_{k-1} \oplus \mathbb{M}_k}^{D_k}, P_{M_k}^{D_k}\right) \leq \sqrt{\delta}, \qquad (12)$$

where $\mathcal{D}_{\text{JS}}(\cdot, \cdot)$ denotes JS distance, the arithmetic square root of JS divergence.

**Bounding the Global JS Distance:** To extend the local constraint to the global upper bound, we use the triangle inequality for the JS distance:

$$\mathcal{D}_{\text{JS}}\left(P_{\mathbb{M}}^D, P_{M_k}^D\right) \leq \mathcal{D}_{\text{JS}}\left(P_{\mathbb{M}}^D, P_{M_{k-1} \oplus \mathbb{M}_k}^D\right) + \\ \mathcal{D}_{\text{JS}}\left(P_{M_{k-1} \oplus \mathbb{M}_k}^D, P_{M_k}^D\right) \qquad (13)$$

The memory $M_{k-1}$ is a subset of $M_{k-1} \oplus \mathbb{M}_k$, which, in turn, is a subset of $\mathbb{M}$. Thus, $M_{k-1} \oplus \mathbb{M}_k$ more closely resembles the original full memory $\mathbb{M}$. Consequently, the probability distribution $P_{M_{k-1} \oplus \mathbb{M}_k}^D$, conditioned on $M_{k-1} \oplus \mathbb{M}_k$, is more similar to $P_{\mathbb{M}}^D$ than to $P_{M_{k-1}}^D$:

$$\mathcal{D}_{\text{JS}}\left(P_{\mathbb{M}}^D, P_{M_{k-1} \oplus \mathbb{M}_k}^D\right) \leq \mathcal{D}_{\text{JS}}\left(P_{\mathbb{M}}^D, P_{M_{k-1}}^D\right). \qquad (14)$$

In other words, adding more chunks from the full memory reduces the divergence with $P_{\mathbb{M}}^D$. Moreover, because the demonstrations $D_k$ are most closely related to the context in the $k$-th chunk itself, we empirically assume:

$$\mathcal{D}_{\text{JS}}\left(P_{M_{k-1} \oplus \mathbb{M}_k}^D, P_{M_k}^D\right) \leq \mathcal{D}_{\text{JS}}\left(P_{M_{k-1} \oplus \mathbb{M}_k}^{D_k}, P_{M_k}^{D_k}\right) \qquad (15)$$

**Theoretical Upper Bound:** Combining the Equations (12) to (15), we can derive the following global upper bound of the information loss:

$$\mathcal{D}_{\text{JS}}\left(P_{\mathbb{M}}^D, P_{M_K}^D\right) \leq (K-1)\sqrt{\delta} + \epsilon, \qquad (16)$$

where the $\epsilon$ is the JS distance between the probability distribution of full context $\mathbb{C}$ and that of the first uncompressed chunk $\mathbb{C}_1$. Hence, by applying the local constraint at each chunk-compression step, the global JS distance $\sqrt{\Delta}$ between the compressed memory and the full memory is controlled by $\delta$ and chunk number $K$. More proof and analysis are depicted in Appendix D.

## 4. Experiments

### 4.1. Experimental Setting

**Evaluation Dataset**. We evaluate our EMLoC on six challenging benchmarks: *ImageNet100*, a subset of ImageNet-1k (Deng et al., 2009) with the first 100 classes for recognition, *ScreenSpot* for cross-platform GUI grounding, *MME-RW* for real-world multimodal tasks, *IllusionVQA* for illusion understanding, *OK-VQA* for knowledge-based QA, and *YouCook2* for video understanding. For datasets without predefined validation splits, we randomly sample 100 test examples for evaluation.

*Table 1.* Results of efficient multi-modal long context learning (EMLoC) in various downstream tasks. The value in the gray cell is the context length. † denotes using 50 examples. ‡ represents utilizing 200 examples.

| Method | Example Number | ImageNet100 | ScreenSpot | MME-RW | IllusionVQA | OK-VQA | YouCook2 |
|---|---|---|---|---|---|---|---|
| Llava1.5 (7B) | | 12.3 | 9.7 | 28.2 | 24.1 | 53.6 | - |
| InternVL2 (8B) | | 12.5 | 2.7 | 33.9 | 28.0 | 47.1 | 88.0 |
| Llama3.2-V (11B) | 0 | 47.6 | 8.1 | 14.6 | 33.0 | - | - |
| MiniCPM-V2.6 (8B) | | 31.0 | 0.3 | 37.2 | 34.6 | 48.3 | 3.3 |
| Qwen2-VL (7B) | | 28.0 | 14.2 | 36.6 | 35.3 | 52.1 | 25.4 |
| MLoC (Qwen2-VL) | 5 | 43.2 † | 14.7 | 39.4 | 38.8 | 58.4 | 86.9 |
| | | 4109 | 1996 | 1924 | 1826 | 1401 | 5907 |
| | 20 | 62.6‡ | 18.2 | 41.1 | **40.9** | 58.6 | **108.8** |
| | | 16264 | 7905 | 7393 | 7271 | 5730 | 23464 |
| EMLoC (Qwen2-VL) | 20 | **63.6** ‡ | **18.3** | **42.2** | **40.9** | **58.7** | 102.0 |
| | | 3643 | 1415 | 1510 | 1878 | 934 | 6218 |

**Implementation Details.** Experiments are conducted on NVIDIA L20 GPUs with 48GB of memory. Inference time is measured with a batch size of 1 on one GPU. The default JS divergence threshold $\delta$ is set to 0.005, and the chunk size is 1.6k. The retention ratio set $R$ is [0.1, 0.2, 0.5, 1.0]. We use Qwen2-VL as the base model due to its support for 32k multi-modal long contexts in any format and a vision encoder that processes images at any resolution without tiling. For the ImageNet100 dataset, the image resolution is 224×224, while other benchmarks use 448×448. The evaluation of other MLLMs uses their default image resolution.

### 4.2. Experimental Results

**Performance of EMLoC on multiple tasks.** As shown in Table 1, multi-modal long-context learning significantly enhances Qwen2-VL's performance across all downstream tasks, whether using 5 or 20 demonstration examples. In tasks like ImageNet100 and YouCook2, leveraging an extremely long context yields substantial performance gains.

These results underscore the importance of efficient multi-modal long context learning for practical applications, particularly in ImageNet100 and YouCook2. When utilizing 20 examples, our EMLoC dramatically reduces the average context length from *11338* to *2600*, a remarkable 77% reduction, without sacrificing performance. Notably, our EMLoC outperforms MLoC with 5 demonstration examples and *3530* average context length by a wide margin, even with a shorter context length. Furthermore, when utilizing 20 examples, EMLoC surpasses MLoC with full memory in most benchmarks. This improvement is likely due to the removal of irrelevant background noise or distractions during compression. As a result, the compressed memory is more precise, leading to better performance.

Meanwhile, inference costs have been significantly reduced. On ImageNet100 with 200 examples, EMLoC reduces *inference FLOPs* from 1.76T to 1.35T and total *inference time* from 1866s to 1107s compared to MLoC.

*Table 2.* Multi-modal long context learning with varying numbers of examples on ImageNet 100. The value in the gray cell is the context length.

| Method | Number of Examples | | | | |
|---|---|---|---|---|---|
| | 0 | 25 | 50 | 200 | 300 |
| MLoC | 28.0 | 39.9 | 43.2 | 62.6 | 62.9 |
| | 0 | 2053 | 4109 | 16264 | 24468 |
| EMLoC | 28.0 | 39.4 | 46.2 | 63.7 | 62.6 |
| | 0 | 565 | 946 | 3643 | 5365 |

**Results with varying numbers of examples.** Table 2 shows how the performance of MLoC and EMLoC changes as the number of examples increases on the ImageNet100 dataset. Both MLoC and EMLoC show steady improvement with more examples, demonstrating strong long-context learning capabilities. However, this improvement comes with a sharp rise in context length, which inflates computational overhead. In contrast, EMLoC compresses the context by nearly a quarter while maintaining or even surpassing MLoC's accuracy (e.g., 63.7 vs. 62.6 with 200 examples). This significant reduction in context length substantially decreases inference time while preserving performance.

**Comparison with other multi-modal ICL methods.** We have compared EMLoC with two other multi-modal in-context learning methods, RICES (Alayrac et al., 2022) in Flamingo and MTV (Huang et al., 2024a), on ImageNet100, MME-RW, and OK-VQA in Table 4.2. RICES retrieves the top 25% most relevant in-context samples from all samples. MTV extracts the mean activation of in-context examples as task vectors and finds the optimal replacement position of these task vectors. During inference, MTV replaces these task vectors at the optimal position of the test sample, which fails to facilitate these tasks. Our EMLoC achieves better average performance across the three benchmarks. It's worth noting that RICES is an online retrieval-augmented method, so it needs to forward the retrieved long context during each inference step. *RICES takes 5 hours inference time and*

*Table 3.* Comparison with other multi-modal in-context learning (ICL) methods.

| Method | ImageNet100 | MME-RW | OK-VQA |
|---|---|---|---|
| MTV | 32.7 | 27.8 | - |
| RICES | **64.5** | 40.5 | 58.5 |
| EMLoC | 63.7 | **42.2** | **58.7** |

*Table 4.* Comparison with fine-tuning methods on ImageNet100 with 200 examples, MME-RW with 20 examples, and OK-VQA with 20 examples.

| Method | ImageNet100 | MME-RW | OK-VQA | Average |
|---|---|---|---|---|
| $M^2PT$ | **65.2** | 31.6 | 15.6 | 37.5 |
| VPT | 43.6 | 38.7 | 54.5 | 45.6 |
| $E^2PT$ | 48.6 | 39.0 | 55.8 | 47.8 |
| Full Fine-tuning | 64.7 | **42.7** | 49.7 | 52.4 |
| LoRA | 61.1 | 42.1 | **60.9** | 54.7 |
| EMLoC | 63.7 | 42.2 | 58.7 | **54.9** |

*43G memory cost on ImageNet100, while our EMLoC requires only 18 minutes with 18G memory*, showing clear advantages in efficiency.

**Comparison with fine-tuning methods.** EMLoC is a training-free adaptation method that does not update any parameters of the MLLM. With less adaptation time (144s), EMLoC is comparable to LoRA (234s) (Hu et al., 2022) and full fine-tuning (820s). Our EMLoC also surpasses other PEFT methods, such as $M^2PT$ (Wang et al., 2024c), VPT (Jia et al., 2022), and $E^2PT$ (Han et al., 2023a). The training details of these PEFT methods and full fine-tuning are depicted in Appendix E.1.

### 4.3. Ablation Studies

#### Comparison with other pruning strategies.

The layer-adaptive pruning strategy employs JS divergence analysis to dynamically evaluate token importance across layers. Unlike fixed pruning approaches, EMLoC maintains superior performance through selective token preservation. As evidenced in Table 5, at 22.4% compression on ImageNet100, EMLoC sustains 63.7 while competitors suffer significant degradation. This demonstrates our method's effectiveness in identifying and retaining critical tokens. The performance boundary emerges at extreme compression (15.7%, $\delta$=0.02), where EMLoC's accuracy drops to 57.7, indicating essential token removal, while conventional methods show paradoxical stability (47.3-48.2). This suggests competing approaches fail to properly distinguish crucial tokens even at moderate compression (22.4%), as their performance collapses before reaching critical pruning thresholds. A more comprehensive comparison with additional pruning strategies across various metrics can be found in

*Table 5.* Comparison with other pruning strategies under the different compression ratios on ImageNet100 with 200 demonstrations and MME-RW with 20 demonstrations

| Method | ImageNet100 | | MME-RW | |
|---|---|---|---|---|
| | Ratio | Acc | Ratio | Acc |
| MLCL | 100% | 62.6 | 100% | 41.1 |
| SnapKV/H2O | 22.4% | 47.6 | 20.4% | 39.6 |
| PyramidKV | 22.4% | 49.3 | 20.4% | 39.8 |
| EMLoC(Ours) | 22.4% | **63.7** | 20.4% | **42.2** |
| SnapKV/H2O | 15.7% | 47.3 | 14.1% | 40.2 |
| PyramidKV | 15.7% | 48.2 | 14.1% | 39.4 |
| EMLoC(Ours) | 15.7% | **57.7** | 14.1% | **40.6** |

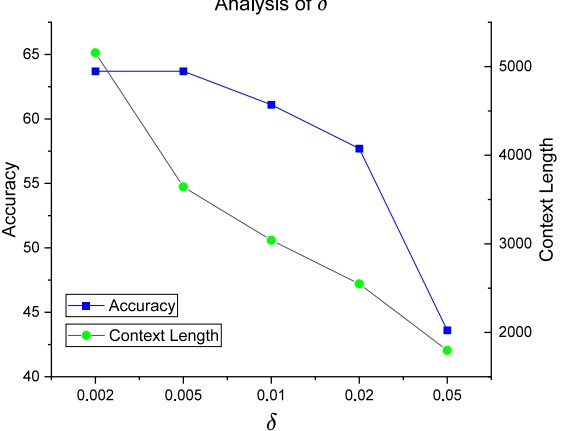

*Figure 3.* Performance and context length trends of EMLoC on ImageNet100 with 200 examples across different $\delta$ values

Appendix C.1.

**Results of different $\delta$.** Figure 3 presents the effect of varying the Jensen-Shannon divergence threshold. A higher threshold results in greater compression but also leads to increased information loss. As the threshold increases, both accuracy and context length decrease. It demonstrates that $\delta$ can effectively control the global upper bound $\Delta$. The default threshold of 0.005 is selected as a balance between compression efficiency and performance preservation.

**Results in various chunk lengths.** Table 6 examines chunk length impacts on computational efficiency and compression effectiveness. Longer chunks linearly increase memory demands and compression ratios while maintaining stable adaptation times (150s) and accuracy (62.4-63.7). Our chunk-wise compression enables memory-efficient processing without performance degradation - notably supporting consumer GPUs (NVIDIA 4090) at 0.8k chunks with merely 1.3 accuracy drop. The default 1.6k configuration balances memory usage (38G) and compression effectiveness (22.4 ratio), while extreme 3.2k lengths cause OOM errors.

*Table 6.* Results in various chunk lengths and chunk numbers on ImageNet100 with 200 demonstrations.

| Chunk Length | Chunk Number | Adaptation Time | Acc | Peak Memory | Ratio |
|---|---|---|---|---|---|
| 0.8k | 20 | 150s | 62.4 | 24G | 20.4 |
| 1.6k | 10 | 144s | **63.7** | 38G | 22.4 |
| 2.4k | 7 | 149s | 63.4 | 44G | 24.7 |
| 2.8k | 6 | 142s | 62.5 | 48G | 26.3 |
| 3.2k | 5 | - | - | OOM | - |

*Table 7.* Results under different retention ratios.

| Retention Ratios | Context Length | Ratio | Acc |
|---|---|---|---|
| 0.1, 0.2, 0.5, 1.0 | 3643 | 22.4% | **63.7** |
| 0.2, 0.5, 1.0 | 4952 | 30.4% | 61.6 |
| 0.05, 0.1, 0.2, 0.5, 1.0 | 3421 | 21.0% | 58.6 |
| 0.1, 0.2, 0.5, 0.8, 1.0 | 3728 | 22.9% | 62.2 |

**Results with different retention ratios.** The retention ratio $r$ is selected greedily from an ascending predefined retention ratios set $R$, progressing to larger ratios until satisfying the JS divergence constraint or reaching 1.0. Two distinct patterns emerge based on the minimum ratio setting: When significantly below optimal compression levels (e.g., 0.05 vs 22.4% optimal), $\delta$ becomes the dominant control factor. As shown in Table 6 and Figure 3, this configuration enables adaptive balancing - excessive layer compression is offset by others retaining more tokens to meet JS constraints, maintaining comparable overall ratios (21.0% vs 22.4%). Near-optimal minimum ratios (e.g., 0.2) directly determine final compression outcomes (30.4%). Based on empirical observations, we recommend setting the minimum ratio to approximately half of the optimal compression ratio. If the minimum ratio is too small, the overall compression ratio may not be sufficiently reduced, while simultaneously discarding both important and redundant tokens at some layers. The final retention ratios $R$ are set to [0.1, 0.2, 0.5, 1.0].

**Abalation studies observation window.** For each token in the multi-modal long context, we use the answer tokens as the observation window to compute its importance score. The answer tokens are the most critical elements in the Image-Question-Answer pair, as they encapsulate the key semantic information of the example. In Table 8, we compare different observation windows. The results show that answer tokens serve as an effective observation window, while the question tokens and image tokens often include a significant amount of irrelevant or noisy data. Furthermore, as highlighted in the last row of Table 8, retaining answer tokens yields better performance, further confirming that answer tokens are crucial in multi-modal contexts.

*Table 8.* Abalation studies observation window. ∗ indicates preserving all answer tokens in the memory.

| Observation Window | Context Length | Ratio | Acc |
|---|---|---|---|
| Answer | 3643 | 22.4% | 61.9 |
| Question+Answer | 5304 | 32.6% | 58.3 |
| Image+Question+Answer | 8796 | 54.0% | 55.4 |
| Answer∗ | 3643 | 22.4% | **63.7** |

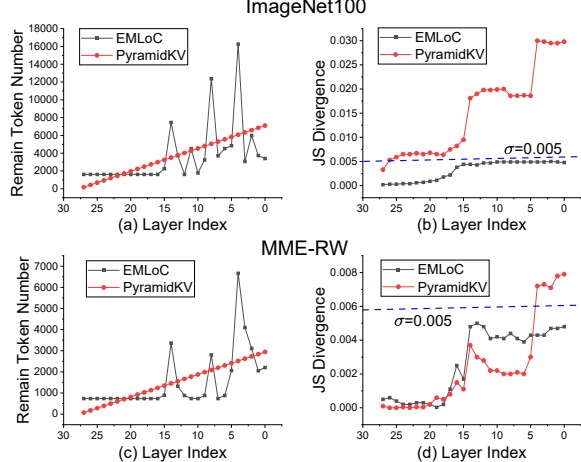

*Figure 4.* Remaining token number of EMLoC and PyramidKV in ImageNet100 with 200 demonstrations and MME-RW with 20 demonstrations. The corresponding JS divergence after pruning is also illustrated to demonstrate the advantage of EMLoC.

### 4.4. Visualization of Compression

**Remaining token number across layers** Previous studies (Xiao et al., 2024; Cai et al., 2024) suggest that earlier layers in large language models (LLMs) are more important than later ones, advocating for a pyramid-shaped pruning strategy. However, our layer-wise adaptive pruning approach challenges this assumption, arguing that layer importance should be determined dynamically based on task-specific demonstrations rather than a fixed heuristic.

As shown in Figure 4, experiments on ImageNet100 and MME-RW reveal that layer importance does not strictly follow a pyramid structure. Instead, certain early layers are particularly critical, as indicated by the spikes in Figure 4(a) and (c). Pruning tokens in these key layers causes significant shifts in the model's output distribution. For instance, in Figure 4(b) and (d), applying PyramidKV pruning to these layers leads to a sharp increase in JS divergence, significantly degrading performance. The corresponding performance metrics are provided in Table 5. Our findings highlight that layers 4, 8, and 14 of Qwen2-VL are crucial in both ImageNet100 and MME-RW, as they retain significantly more tokens than adjacent layers.

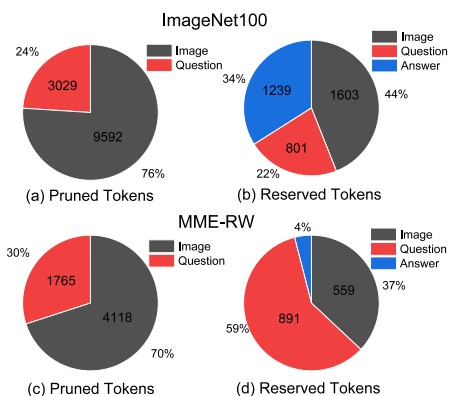

*Figure 5.* Distribution of pruned and reserved tokens.

**Distribution of Pruned and Reserved Tokens.** This study serves as an initial exploration of multi-modal context compression. As shown in Figure 5(a) and (c), image tokens make up the majority of pruned tokens, suggesting that the visual modality contains more redundancy than the textual modality. Besides, Figure 5(b) and (d) indicate that fewer image tokens are retained compared to text tokens, demonstrating effective pruning of redundant visual information. Additionally, variations in tasks may lead to differences in their compression rates.

## 5. Conclusion

This paper presents EMLoC, a training-free method combining chunk-wise compression with layer-wise adaptive pruning to build a compact, task-specific memory for downstream tasks. Experiment results show EMLoC reduces inference overhead while preserving strong performance in multiple vision-language tasks, providing a scalable solution for efficient multi-modal long context learning.

## Acknowledgements

This work is supported in part by Grant No. 2023-JCJQ-LA-001-088, in part by the Natural Science Foundation of China under Grant No. U20B2052, 61936011, 62236006, in part by the Okawa Foundation Research Award, in part by the Ant Group Research Fund, and in part by the Kunpeng&Ascend Center of Excellence, Peking University.

## Impact Statement

This paper presents work whose goal is to advance the field of Machine Learning. There are many potential societal consequences of our work, none which we feel must be specifically highlighted here.

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

# A. Pseudo Code of Layer-wise Adaptive Pruning

---

**Algorithm 1:** Layer-wise Adaptive Pruning

---

**Input:** $\mathbb{C}_k \in \mathbb{R}^S$: concatenated context of $k$-th chunk, where $S = \frac{N}{K} \times T$;

$D_k \in \mathbb{R}^{\frac{N}{K} \times T}$: demonstration examples of $k$-th chunk in a batch;

$M_{k-1}$: KV cache of previous $k-1$ chunks;

**Output:** $M_k$: Compressed memory of $k$ chunks.

$\mathbb{M}_k \leftarrow \text{ExtractKV}(M_{k-1}, \mathbb{C}_k)$;

$p_{\text{ori}}, \alpha, H \leftarrow \text{MLLM}(M_{k-1} \oplus \mathbb{M}_k, D_k)$;

$\hat{\mathbb{M}}_k \leftarrow \mathbb{M}_k$;

**for** $l \leftarrow L$ **to** $1$ **do**

$\quad \beta^l \leftarrow \left\{ \sum_{i \in \text{ans\_index}} \alpha_{ij}^l \right\}_{j=1}^S$;

$\quad$ **for** $r \in R$;          // $R$ is retention ratio set in ascending order

$\quad$ **do**

$\quad\quad \hat{\mathbb{M}}_k^l \leftarrow \mathbb{M}_k^l \left( \text{Topk}(\beta^l, r \times S) \right)$;

$\quad\quad$ ;                               // $S$ is chunk length

$\quad\quad p_{\text{iter}} \leftarrow \text{MLLM}(M_{k-1}^{\geq l} \oplus \hat{\mathbb{M}}_k^{\geq l}, H^l)$;

$\quad\quad \ell \leftarrow \text{JS}(p_{\text{ori}}, p_{\text{iter}})$;

$\quad\quad$ **if** $\ell \leq \delta$ **then**

$\quad\quad\quad$ **break**;

$\quad\quad\quad$ ;                          // $\delta$: JS divergence threshold

$M_k \leftarrow M_{k-1} \oplus \hat{\mathbb{M}}_k$;

---

# B. Prompt Template

In ImageNet100, 200 multi-modal examples are evenly divided into 10 chunks. Each image (224×224) is encoded into approximately 64 tokens (may vary slightly due to dynamic aspect ratios), and the corresponding question-answer pair adds around 20 tokens, resulting in about 80 tokens per example. Each chunk contains 20 examples (roughly 1.6k tokens). The system prompt appears only at the start of the first chunk. Below is an example structure:

```
# Start of 1st chunk
<|im_start|> system\n You are a helpful assistant.<|im_end|>
## sample 1
<|im_start|> user\n <|vision_start|> <Image1.jpg> <|vision_end|> What category
does the image belong to?  <|im_end|>
<|im_start|> assistant\n <class 1>.  <|im_end|> ...
# Start of 2nd chunk
## sample 21
<|im_start|> user\n <|vision_start|> <Image21.jpg> <|vision_end|> What category
does the image belong to?  <|im_end|>
<|im_start|> assistant\n <class 11>.  <|im_end|> ...
```

For other image benchmarks, each image is encoded into 256 tokens (448×448 resolution). Each chunk has 4 examples, resulting in a chunk size of 1.1k–1.6k tokens. For the YouCook2 video benchmark, each video with 8 frames is encoded into 1024 tokens, with 4 videos per chunk, yielding a 4.7k chunk size. If sample lengths vary significantly, we use a greedy algorithm to progressively fill each chunk up to a maximum size.

*Table 9.* Comparison with other context compression methods on ImageNet100. EMLoC* increase the chunk number from 10 to 20 and utilize a group-wise strategy to save adaptation memory and time.

| Method | Retention Ratio | Adapt Time | Adapt Memory | Infer Time | Infer Memory | Acc |
|---|---|---|---|---|---|---|
| MLoC | 100% | 28s | 62G | 31m | 19G | 62.6 |
| PyramidKV | 22.4% | 54s | 34G | 19m | 17G | 49.3 |
| FastGen | 36.0% | 45s | 38G | 37m | 21G | 49.3 |
| PyramidInfer | 24.6% | 41s | 42G | 21m | 17G | 55.6 |
| EMLoC | **22.4%** | 144s | 38G | **18m** | **17G** | **63.7** |
| EMLoC* | 27.6% | 85s | **24G** | 19m | **17G** | 60.9 |

*Table 10.* Comparison with LongVA and MLoC on VideoMME w/o subtitles with 384 frames.

| Method | Context Length | LLM FLOPs | LLM Time | Peak Memory | Overall ACC |
|---|---|---|---|---|---|
| LongVA | 55.5k | 1715.5T | 22h | 41G | 51.8 |
| MLoC | 27.9k | 554.8T | 7h | 38G | **60.3** |
| EMLoC | **2.3k** | **272.0T** | **5h** | **24G** | 60.1 |

# C. More Experiments

## C.1. Comparison with other Pruning Strategies

In Section 4.3 and Table 5 of the original paper, we compared our adaptive EMLoC with two static KV-cache algorithms. Table 9 extends this comparison (Table 5) by including PyramidInfer (Yang et al., 2024) and FastGen (Ge et al., 2023). Most KV-cache methods focus on uni-modal text compression, but fail to maintain original performance with a high compression ratio. EMLoC retains only 22.4% of tokens while achieving 63.7% accuracy, outperforming FastGen (49.3% accuracy with 36% tokens) and PyramidInfer (55.6% accuracy with 24.6% tokens). Unlike existing KV-cache methods, EMLoC effectively maintains the full-context performance while significantly reducing the context length, thus improving efficiency.

To optimize the trade-off between adaptation cost and inference performance, we explore *increasing the chunk number from 10 to 20 and a group-wise strategy* (every two layers share the same retention ratio). This variant, *EMLoC*\*, reduces adaptation time from 144s to 85s and memory from 38G to 24G, at the cost of a slight accuracy degradation($63.7 \rightarrow 60.9$) and a higher retention ratio ($22.4\% \rightarrow 27.6$). This allows for a flexible implementation in computation constrained scenarios. The adaptation cost is significantly smaller compared with its gains in inference efficiency.

## C.2. Experiment on Long-Video Benchmark.

In LongVA, each frame consists of 144 tokens, whereas in Qwen2-VL, 144 tokens represent every two frames through temporal pooling. Compared to our baseline MLoC, *EMLoC significantly reduces computational overhead while maintaining nearly the same accuracy.* Specifically, EMLoC reduces the average context length from 27.9k to just 2.3k tokens, LLM FLOPs from 554.8T to 272.0T, inference time from 7 hours to 5 hours, and peak GPU memory from 38G to 24G, while preserving a consistent accuracy (60.1 vs. 60.3).

To achieve this efficiency, we set $\delta = 0.04$ and configure the retention ratio to: [0.02, 0.1, 0.5, 1.0]. Instead of optimizing the retention ratio for each layer individually (layer-wise), we adopt a *group-wise strategy*, where every 14 layers are treated as a single group and share the same retention ratio. This allows for a more stable and efficient selection process during online inference. Under an identical setup (384 frames at the same resolution), both MLoC and EMLoC outperform LongVA while requiring significantly fewer computations. EMLoC also enables real-time long-video understanding on consumer-grade GPUs such as the NVIDIA 3090, making it a more practical solution for real-world applications.

## C.3. Robustness of Hyper-parameters

Those parameters have clear meanings and are easy to adjust. For a high compression ratio, we can set a smaller retention ratio and a higher JS threshold $\delta$, and the optimal pruning strategy will be identified heuristically. Our method avoids manually adjusting numerous parameters like FastGen (Ge et al., 2023) or PyramidInfer (Yang et al., 2024). Our experiments also show that the default hyperparameters are stable across different tasks, as depicted in Table 11 and Table 12.

*Table 11.* Robustness of JS Threshold $\delta$. We show the performance under different $\delta$ values across tasks

| $\delta$ | ImageNet100 | MME-RW | OK-VQA |
|---|---|---|---|
| 0.002 | 63.7 | 42.2 | 58.6 |
| **0.005** | **63.7** | **42.2** | **58.7** |
| 0.02 | 57.7 | 41.0 | 57.0 |

*Table 12.* Robustness of retention ratios across different tasks. We provide the performance under different retention ratios across tasks to demonstrate the robustness.

| Retention Ratios | ImageNet100 | MME-RW | OK-VQA |
|---|---|---|---|
| [0.05, 0.1, 0.2, 0.5, 1] | 58.6 | 41.6 | 58.3 |
| **[0.1, 0.2, 0.5, 1]** | **63.7** | **42.2** | **58.7** |
| [0.2, 0.5, 1] | 61.6 | 41.7 | 58.6 |

## C.4. Impact of task disparity on EMLoC

The impact measurement follows the method in (Han et al., 2024), with some modifications. Given that Qwen2VL(MLLM) is pretrained on millions of image-text pairs covering lots of vision-language tasks, we use zero-shot accuracy as an indicator of the task disparity between downstream datasets and the pretraining datasets.

MLLMs perform well on OK-VQA (52.1%), suggesting that the data and task of OK-VQA are highly similar to those seen during pretraining. Meanwhile, ImageNet100 achieves 28% accuracy, indicating moderate similarity. In contrast, MedXpertQA only reaches 21.5% accuracy—near random chance in a five-choice QA—indicating significant dissimilarity.

Based on Table 13 and Table 14, the impact of task disparity can be summarized as follows:

- Low task disparity (OK-VQA, ImageNet100): When task disparity is low(highly or moderately similar to pretrained data), our EMLoC adapts well to the downstream tasks. It outperforms full fine-tuning on OK-VQA when data is limited, and achieves a average accuracy 48.2%, which is comparable to LoRA's 47.8%.

- High task disparity & scarce data (MedXpertQA): All methods struggle. Adapting to truly novel tasks typically demands extensive continued pretraining or finetuning (Lu et al., 2025).

- Larger downstream datasets(ImageNet100 with 200 examples): Full fine-tuning slightly outperforms both LoRA and EMLoC, echoing (Han et al., 2024)'s finding that "full fine-tuning gradually closes the performance gap as dataset size grows."

- Other tasks with low disparity(see Table 1): EMLoC also facilitates on tasks with small task disparity, such as MME-RW (OCR, remote sensing, driving), IllusionVQA (optical illusions), and YouCook2 (video captioning/activity recognition).

*Table 13.* Task disparity between downstream datasets and pretrained datasets.

| Dataset | Zero-Shot Accuracy | Task | Similarity to Pretrained Dataset | Task Disparity |
|---|---|---|---|---|
| OK-VQA | 52.1 | common-sense QA | Highly similar | Low |
| ImageNet100 | 28.0 | image classification | Moderately similar | Low |
| MedXpertQA | 21.5 (near random) | medical QA (medical image) | Dissimilar | High |

## D. Analysis of $\Delta$

Our theoretical analysis establishes a linear relationship between the global JS distance $\sqrt{\Delta}$ and the chunk count $K$ under worst-case assumptions (Equation (16)). However, empirical observations reveal more nuanced behavior. As shown in Figure 6(a), $\Delta$ exhibits a strong positive correlation with the local constraint $\delta$, aligning with our theoretical prediction that $\sqrt{\Delta} \propto \sqrt{\delta}$. This confirms that $\delta$ effectively governs the global information loss, enabling practitioners to reliably control compression quality through this single parameter.

*Table 14.* Impact of task disparity on different adaptation methods.

| Method | OK-VQA | ImageNet100 | MedXpertQA | Average |
|---|---|---|---|---|
| Baseline (Qwen2-VL) | 52.1 | 28.0 | 21.5 | 33.9 |
| LoRA | **60.9** | 61.1 | 21.5 | 47.8 |
| Full Fine-tuning | 49.7 | **64.7** | 22.0 | 45.5 |
| EMLoC | 58.7 | 63.7 | **22.2** | **48.2** |

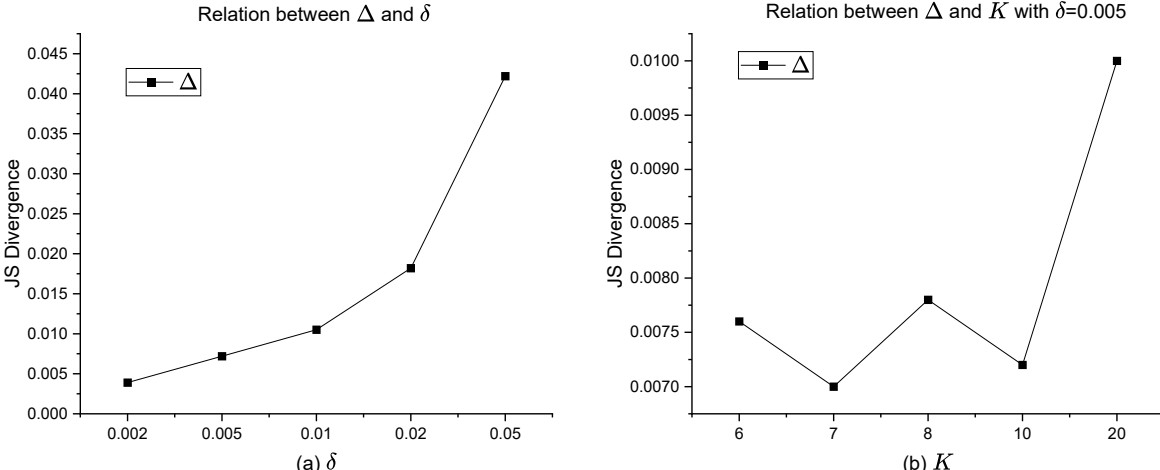

*Figure 6.* Trend of $\Delta$ on various $\delta$ and $K$.

Notably, Figure 6(b) demonstrates that $\Delta$ remains stable within a narrow range across varying $K$ values when $\delta$ is fixed. This diverges from the theoretical upper bound's linear dependence on $K$, suggesting our worst-case analysis accommodates challenging scenarios where chunk dependencies might compound errors. In practice, however, the weak inter-chunk dependencies in real-world datasets prevent error accumulation across compression steps. Consequently, the effective upper bound simplifies to $\Delta \leq \gamma\delta$, where $\gamma$ is a small constant (typically $\gamma \leq 2$ in our experiments), rather than scaling with $K$.

This phenomenon has important practical implications:

- Hyperparameter tuning focuses primarily on $\delta$, substantially reducing configuration complexity

- Users can freely increase $K$ to minimize memory usage without compromising information integrity

Our method thus achieves an optimal balance between theoretical rigor and practical usability - while the theoretical bound guarantees robustness, the empirical independence between chunks enables memory-efficient compression through large $K$ values. This dual advantage makes our approach particularly suitable for long-context applications where GPU memory constraints are critical.

## E. More Implementation Details

### E.1. Training Details

In Table 4, we compare EMLoC with some fine-tuning methods, such as LoRA (Hu et al., 2022), fine-tuning, M$^2$PT (Wang et al., 2024c), VPT (Jia et al., 2022), and E$^2$PT (Han et al., 2023a).

In LoRA adaptation, we apply LoRA adapters to all linear modules of the LLM, including qkv_proj, out_proj, up_proj, and down_proj, while keeping the vision encoder and multi-modal projector frozen. The rank and alpha are set to 16 and 32, respectively. In full fine-tuning, only the LLM is fine-tuned with DeepSpeed ZeRO-3, leaving other parameters frozen. Other unspecified settings follow the default configurations in LLaMAFactory. The detailed hyperparameters are reported in

Table 15 and Table 16.

*Table 15.* Hyperparameters for LoRA training

| Hyperparameter | Value |
|---|---|
| Optimizer | AdamW |
| learning rate | 3e-5 |
| batch size | 8 |
| warmup ratio | 0.1 |
| epochs | 5 |
| clip norm | 1.0 |

*Table 16.* Hyperparameters for full fine-tuning

| Hyperparameter | Value |
|---|---|
| Optimizer | AdamW |
| learning rate | 1e-5 |
| batch size | 8 |
| warmup ratio | 0.1 |
| epochs | 1 |
| clip norm | 1.0 |

For other PEFT methods, following the default configuration in M²PT (Wang et al., 2024c), the number of visual prompts and textual prompts is 20 and 10, respectively. The learning rate is set to 7e-4. The number of KV prompt token is 5. For Imagenet100, we optimize 5 epochs with 125 steps. For MME-RW and OK-VQA, we just fine-tune 25 steps.

### E.2. Dataset Details

We evaluate our method on *ImageNet100*, a subset of ImageNet-1k (Deng et al., 2009) with the first 100 classes for inference efficiency. Demonstration examples are uniformly sampled from the training set, ensuring even distribution per class. For instance, in the 200-example setting, each class contributes two examples. Evaluation is conducted on the full validation set with 5000 images. Additionally, we evaluate on several other benchmarks: *ScreenSpot*(Cheng et al., 2024) for GUI grounding across diverse platforms, *MME-RW*(Zhang et al., 2024c) for real-world tasks such as OCR, remote sensing, and autonomous driving, *IllusionvQA*(Shahgir et al., 2024) for evaluating optical-illusion understanding, *OK-VQA*(Marino et al., 2019) for open-ended question answering using external knowledge, and *YouCook2* (Zhou et al., 2018) for cooking-video-related tasks, e.g., captioning or activity recognition.

