# OpenReview forum: "Efficient Multi-modal Long Context Learning for Training-free Adaptation"
_ICML.cc/2025/Conference — ICML 2025 poster_

### Official Review · Reviewer_iyMg · 2025-02-24

**Overall Recommendation:** 3

**Summary:**

While current popular adaptation for MLLMs heavily replies on fine-tuning, the paper proposes a novel training-free alternative that can embed demonstration examples directly into the model input. Due to the lengthy inputs might bring computational and memory overhead, the proposed method contributes a chunk-wise compression with layer-wise adaptive pruning.

The proposed method is able to reach a dramatic reduction in inference complexity while retaining the performance.

**Claims And Evidence:**

The angle of this paper is interesting, while I have several questions:

1. In Sec. 3.1, in order to adapt a pre-trained MLLM without any training or parameter fine-tuning, the proposed approach construct a long context by concatenating task-specific demonstration examples. This raise my first question on the disparity between the pre-trained MLLM dataset and adaptation tasks. In [1], the paper demonstrates that task disparity might be a huge impact on vision task adaptation, would this be the case for in-context learning of MLLM? The authors need to specifically discuss the datasets and possible disparities.

[1] Facing the Elephant in the Room: Visual Prompt Tuning or Full Finetuning? ICLR 2024.

**Essential References Not Discussed:**

Multimodal parameter finetuning papers, such as M2PT: Multimodal Prompt Tuning for Zero-shot Instruction Learning. EMNLP 2024 is not discussed. Other PEFT methods, though not common under the multimodal settings, should be covered/discussed as well.

**Experimental Designs Or Analyses:**

The experimental designs are current insufficient, please see "Methods And Evaluation Criteria". More parameter-efficient fine-tuning approaches should be discussed, no simply having LoRA and full fine-tuning.

**Methods And Evaluation Criteria:**

The main table needs further improvements. The current version only includes baselines such as MLoC with given number of examples. The proposed EMLoC, however, does not have a systematic comparison to fine-tuning methods (I have acknowledged that in Table 3 the authors included LoRA, and Full fine-tuning for comparison). However, more PEFT fine-tuning methods [1-5] should be included for completeness.

Right now, I could not see the huge advantage that EMLoC can bring. This is very critical, as the authors claim that the proposed method is a good alternative to current fine-tuning approaches (including full fine-tuning and other fine-tuning approaches).

[1] Visual Prompt Tuning. ECCV 2022.

[2] E2VPT: An Effective and Efficient Approach for Visual Prompt Tuning. ICCV 2023.

[3] Adaptformer: Adapting vision transformers for scalable visual recognition. NeurIPS 2022.

[4] Learning expressive prompting with residuals for vision transformers. CVPR 2023.

[5] M2PT: Multimodal Prompt Tuning for Zero-shot Instruction Learning. EMNLP 2024.

**Other Comments Or Suggestions:**

N/A

**Other Strengths And Weaknesses:**

N/A

**Questions For Authors:**

The paper is interesting, paving a promising way for leveraging the in-context learning for MLLM new task adaptation. However, there are two fundamental problems in current paper:

1. See "Claims And Evidence";

2. See "Methods And Evaluation Criteria".

**Relation To Broader Scientific Literature:**

The method proposed in this paper is interesting, instead of exhaustively digging from the area of finetuning, the method leverages the power of in-context learning for new task adaptation.

**Theoretical Claims:**

The theoretical claims of this paper is easy to follow, and I do not see any issues during review.

---

> ### Author Rebuttal · Authors · 2025-03-31
>
> **Q1:** Disparity between the pre-trained MLLM dataset and adaptation tasks. In [1], the paper demonstrates that task disparity might have a huge impact on vision task adaptation; would this be the case for in-context learning of MLLM?
>
> **A1:** We appreciate the reviewer’s insightful question on task disparity in MLLM adaptation. The great work [1] highlights the impact of task disparity on vision adaptation, which inspires us to explore influence of task disparity on in-context learning of MLLMs. To evaluate this, we conducted experiments on ImageNet100, OK-VQA, and MedXpertQA.
>
> **Table R4.1: Impact of task disparity on EMLoC**
>
> | Method             | ImageNet100 | OK-VQA | MedXpertQA |
> | :----------------- | :---------- | :----- | :--------- |
> | Baseline(Qwen2-VL) | 43.2        | 52.1   | 21.5       |
> | LoRA               | 61.1        | 60.9   | 21.5       |
> | Full Fine-tuning   | 64.7        | 49.7   | 22.0       |
> | EMLoC              | 63.7        | 58.7   | 22.2       |
>
> For ImageNet100 and OK-VQA, where the MLLM has seen similar data and tasks during pre-training, EMLoC effectively adapts with limited data (200 and 20 examples, respectively). Full fine-tuning struggles on OK-VQA, probably due to the overfitting with only 20 examples. In contrast, all methods fail on MedXpertQA, a complex multi-choice medical dataset, where the baseline model shows poor performance, indicating no prior knowledge in this domain. With only 20 examples, adaptation remains ineffective. These results suggest that EMLoC could  efficiently adapt pre-trained knowledge with minimal data, but struggles when entirely new capabilities are required. This finding aligns with the conclusion in [1]. When there is a significant discrepancy between the pre-trained and downstream tasks, adaptation can be considerably more difficult.
>
> **Q2:** More PEFT fine-tuning methods should be included for completeness.
>
> **A2:** We appreciate the suggestion. In response, we compare **EMLoC** with **VPT** [1], **E²PT** [2], and **M²PT** [5] across three multi-modal tasks.
>
> On ImageNet100 with 200 training examples, M²PT achieves the best performance due to its strong adaptation capacity, leveraging three different adapters(textual/visual prompts and multi-modal projector). However, when the number of training samples is reduced to 20 (as in MME-RW and OK-VQA), M²PT tends to overfit due to its increased number of optimized parameters. VPT, which optimizes only the visual prompts of the visual encoder, has fewer trainable parameters and a lower optimization capacity, leading to a weaker performance across all three tasks. However, its limited parameter tuning helps to mitigate overfitting risks to some extent. With visual prompt tokens and the powerful shared KV prompt tokens,  E$^2$PT achieves better fine-tuning performance than VPT. Following M²PT, the number of visual prompts and textual prompts are 20 and 10, respectively. The learning rate is set to 7e-4. The number of KV prompt token is 5. For Imagenet100, we optimize 5 epochs with 125 steps. For MME-RW and OK-VQA, we just fine-tune 25 steps.
>
> **As a training-free method, EMLoC achieves competitive performance across various tasks** and lowers the risk of overfitting. This makes it effective in low-data regimes, where traditional fine-tuning methods may struggle. These methods will be compared in our revised manuscript. We will also **release the code** of these PEFT methods in **LLaMAFactory** for the research community.
>
> **Table R4.2: Comparison with other PEFT methods on multi-modal benchmarks**
>
> | Method | ImageNet100 | MME-RW   | OK-VQA   |
> | ------ | ----------- | -------- | -------- |
> | VPT    | 43.6        | 38.7     | 54.5     |
> | E²PT   | 48.6        | 39.0     | 55.8     |
> | M²PT   | **65.2**    | 31.6     | 15.6     |
> | EMLoC  | 63.7        | **42.2** | **58.7** |
>
> **Q3:** The advantages of our EMLoC.
>
> **A3:** EMLoC is a training-free adaptation method designed for multi-modal long-context learning, effectively leveraging the strong capabilities of pre-trained MLLMs and offering a reduced risk of overfitting. This aligns with the growing trend in the test-time scaling era.
>
> To the best of our knowledge, EMLoC is the first multi-modal KV-cache pruning method to achieve performance comparable to that of fine-tuning methods, while maintaining a high compression ratio. As shown in Table R2, EMLoC surpass the second best method PyramidInfer by 5.3% in accuracy with a less retention ratio. **In addition, EMLoC can also facilitate the online long-video understanding**. In Table R1.2, EMLoC reduces context length from 27.9k to just 2.3k, LLM FLOPs from 554.8T to 272.0T, and inference time from 7 hours to 5 hours, while preserving a consistent accuracy (60.1 vs. 60.3).
>
> Our adaptive pruning strategy provides valuable insights into multi-modal pruning and the importance of different layers in MLLMs in Fig.4 and Fig.5, which may inspire further exploration in multi-modal pruning techniques.

---

> > ### Comment · Reviewer_iyMg · 2025-04-07
> >
> > Thanks for the rebuttal.
> >
> > I have some further question w.r.t the response.
> >
> > 1. How do the authors measure the impact of task disparity? Is it based on [1] or it is a novel approach?
> >
> > 2. What is the different between VPT [1] and M2PT [5] under your setting? As far as I know, they are both prompt tuning techniques focused on single/multi-modalities?

---

> > > ### Author Response · Authors · 2025-04-08
> > >
> > > Thank you once again for your valuable feedback and insightful questions. We sincerely hope that our responses below address your concerns. If you find our revisions satisfactory, we would be truly grateful if you could kindly reconsider your final rating.
> > >
> > > Q4: How do the authors measure the impact of task disparity? Is it based on [1] or it is a novel approach?
> > >
> > > A4: The impact measurement follows the method in [1], with some modifications. Given that Qwen2VL(MLLM) is pretrained on millions of image-text pairs covering lots of vision-language tasks, **we use zero-shot accuracy as an indicator of the task disparity between downstream datasets and the pretraining datasets**.
> > >
> > > **Table R4.3 Task disparity between downstream datasets and pretrained datasets**
> > >
> > > | Dataset     | Zero-Shot Accuracy | Task                      | Similarity to Pretrained Dataset | Task Disparity |
> > > | ----------- | ------------------ | ------------------------- | -------------------------------- | -------------- |
> > > | OK-VQA      | 52.1               | common-sense QA           | Highly similar                   | Low            |
> > > | ImageNet100 | 28.0               | image classification      | Moderately similar               | Low            |
> > > | MedXpertQA  | 21.5 (near random) | medical QA(medical image) | Dissimilar                       | High           |
> > >
> > > MLLMs perform well on OK-VQA (52.1%), suggesting that the data and task of OK-VQA are highly similar to those seen during pretraining. Meanwhile, ImageNet100 achieves 28% accuracy, indicating moderate similarity. In contrast, MedXpertQA only reaches 21.5% accuracy—near random chance in a five-choice QA—indicating significant dissimilarity.
> > >
> > > Based on Table R4.1 & R4.3, the impact of task disparity can be summarized as follows:
> > >
> > > 1. **Low task disparity (OK‑VQA, ImageNet100):** When task disparity is low(highly or moderately similar to pretrained data), our EMLoC adapts well to the downstream tasks. It outperforms full fine-tuning on OK-VQA when data is limited, and achieves a average accuracy 48.2%, which is comparable to LoRA's 47.8%.
> > >
> > > 2. **High task disparity & scarce data (MedXpertQA):** All methods struggle. Adapting to truly novel tasks typically demands extensive continued pretraining or finetuning [6].
> > >
> > > 3. **Larger downstream datasets(ImageNet100 with 200 examples):** Full fine‑tuning slightly outperforms both LoRA and EMLoC, echoing [1]’s finding that “full fine‑tuning gradually closes the performance gap as dataset size grows.”
> > >
> > > 4. **Other tasks with low disparity(see Table 1):** EMLoC also facilitates on tasks with small task disparity, such as MME‑RW (OCR, remote sensing, driving), IllusionVQA (optical illusions) and YouCook2 (video captioning/activity recognition).
> > >
> > >       [6] Fine-tuning large language models for domain adaptation [Nature 2025, npj computational materials]
> > >
> > >
> > >
> > >
> > > Q5: What is the different between VPT [1] and M2PT [5] under your setting? As far as I know, they are both prompt tuning techniques focused on single/multi-modalities?
> > >
> > > **Table R4.4 Details of VPT, M$^2$PT, and VPT***
> > >
> > > | **Method** |                  **Tuned Components**                  | **Parameters** | **Adaptation Capacity** | **Overfitting Risk** |
> > > | :--------: | :----------------------------------------------------: | :------------: | :---------------------: | :------------------: |
> > > |    VPT     |                     Visual prompts                     |      0.8M      |         Limited         |         Low          |
> > > |    VPT*    |         Visual prompts + Multi-modal projector         |     45.4M      |          High           |         High         |
> > > |    M²PT    | Visual prompts+Textual prompts + Multi-modal projector |     46.4M      |          High           |         High         |
> > >
> > > A5: In our setting, **M²PT** inserts 20 visual prompt tokens into each layer of the visual encoder, 10 textual prompt tokens into each layer of the LLM, and also fine-tunes the multi-modal projector that projects visual features into the LLM input space. **VPT** only adds 20 visual prompt tokens to each layer of the visual encoder. **VPT\*** builds on VPT by additionally fine-tuning the multi-modal projector. The detailed comparison of these PEFT methods is presented in Table R4.4.
> > >
> > > Although both VPT and M²PT are prompt tuning techniques, M²PT involves more tunable parameters, offering greater optimization capacity at the cost of a higher risk of overfitting. VPT*, similar to M²PT, performs well on ImageNet100 with 200 examples but struggles on MME-RW and OK-VQA with limited 20 examples.
> > >
> > > **Table R4.2: Comparison with other PEFT methods**
> > >
> > > | **Method** | **ImageNet100** | **MME-RW** | **OK-VQA** |
> > > | ---------- | --------------- | ---------- | ---------- |
> > > | VPT        | 43.6            | 38.7       | 54.5       |
> > > | VPT*       | 61.2            | 35.1       | 34.8       |
> > > | M²PT       | **65.2**        | 31.6       | 15.6       |
> > > | EMLoC      | 63.7            | **42.2**   | **58.7**   |

---

### Official Review · Reviewer_Niy5 · 2025-03-13

**Overall Recommendation:** 2

**Summary:**

This paper introduces EMLoC(Efficient Multimodal Long Context Learning), a training-free method to embed examples directly into the model input. It is implemented via layer-wise adaptive pruning. The authors first separate the context into chunks to prune tokens by importance measured with Jenson-Shannon divergence. The authors also show that their strategy can retain information within a certain upper bound of information loss. Experiments on diverse dataset and ablations are executed showing efficiency of their work.

**Claims And Evidence:**

This paper claims that the layer-wise adaptive pruning strategy under Jenson-Shannon divergence contributes to the chunk-wise compression mechanism, further improving the accuracy and efficiency in the long context problems in adapting multi-modal large language models. There are no visible problems in the claims.

**Essential References Not Discussed:**

All essential related works are well cited and discussed in this paper.

**Experimental Designs Or Analyses:**

How did you train the model parameters for the context compression? Is it just fine-tuning the model for the imagenet classification? The comparison with LoRA and Full-fine-tuning for the adaptation time seems less unconvincing. If you want to claim your method is efficient at least it should be compare with other context compression methods or vanilla MLoC for the time consumption and memory consumption.

**Methods And Evaluation Criteria:**

Overall idea sounds. However, there are too many hyper-parameters (retention ratio, divergence distance threshold) that need to be found heuristically. In addition, such iterative algorithm can save memory usage, I do not think this computationally efficient. For the evaluation, I have no idea why they compare their method with LoRA and Full fine-tuning in Table 3.

**Other Comments Or Suggestions:**

Figure 1 seems less intuitive. Rather than using demonstration example in the X-axis why don't you use context length?

**Other Strengths And Weaknesses:**

Refer to above sections.

**Questions For Authors:**

Refer to above sections.

**Relation To Broader Scientific Literature:**

This paper contributes to chunk-wise compression of long contexts in multimodal large language models. This method might helpful for the efficient long context generation methods (however, I cannot see the fair explanation or the experiment for the efficiency).

**Theoretical Claims:**

They claim that their layerwise token pruning method satisfies the upper bound of information loss related to the divergence distance threshold and the number of chunks.

---

> ### Author Rebuttal · Authors · 2025-03-31
>
> **Q1:** There are too many hyperparameters (retention ratio, JS threshold) that need to be found heuristically.
>
> **A1:** Thanks for the comments. Those parameters have clear meanings and are easy to adjust. For a high compression ratio, we can set a smaller retention ratio and a higher JS threshold $\delta$, and the optimal pruning strategy will be identified heuristically. Our method avoids manually adjusting numerous parameters like FastGen or PyramidInfer. Our experiments also show that the default hyperparameters are stable across different tasks, as seen in following two tables.
>
> **Table R3.1: $\delta$ across tasks**
>
> | $\delta$  | ImageNet100 | MME-RW   | OK-VQA   |
> | --------- | ----------- | -------- | -------- |
> | 0.002     | 63.7        | 42.2     | 58.6     |
> | **0.005** | **63.7**    | **42.2** | **58.7** |
> | 0.02      | 57.7        | 41.0     | 57.0     |
>
> **Table R3.2: retention ratios across tasks**
>
> | Retention Ratios         | ImageNet100 | MME-RW   | OK-VQA   |
> | ------------------------ | ----------- | -------- | -------- |
> | [0.05, 0.1, 0.2, 0.5, 1] | 58.6        | 41.6     | 58.3     |
> | **[0.1, 0.2, 0.5, 1]**   | **63.7**    | **42.2** | **58.7** |
> | [0.2, 0.5, 1]            | 61.6        | 41.7     | 58.6     |
>
> **Q2:** The iterative algorithm can save memory usage, but I do not think this is computationally efficient.
>
> **A2:** While the iterative algorithm in EMLoC introduces a modest adaptation time(~144 s), this cost is **significantly smaller compared with its gains in inference efficiency and performance**. As depicted in L326-328 and Table R2, on ImageNet100, **EMLoC reduces inference time from 31 minutes of MLoC to 18 minutes**, and memory usage from 19G to 17G without accuracy degradation. Compared to the in-context learning method RICES in Table R1.1, EMLoC reduces inference time from 5 hours to 18 minutes and memory from 43G to 17G. Besides, PyramidKV sacrifice 16.1% accuracy to achieve faster adaptation(55s vs 144s).
>
> To further address the computational concerns, EMLoC offers flexibility:
>
> 1. **Group-wise pruning**: By grouping adjacent layers (e.g., 2 layers per group), adaptation time drops to **85 seconds** (a 40% reduction), with a slight accuracy drop.
> 2. **Decoupled adaptation/inference**: Adaptation needs a **one-time cost** per task, while the pruned KV cache may be repetitively used for **thousands of times** during inference.
>
> Meanwhile, as shown in Table R1.2, EMLoC is also an efficient method in online long-video understanding, reducing the total LLM time from 7 hours to 5 hours, and peak GPU memory from 38G to 24G.
>
> **Q3:** Why compare EMLoC with LoRA and full fine-tuning in Table 3. The comparison with LoRA and full fine-tuning for adaptation time seems less convincing. Comparison with other context compression methods and MLoC for time and memory consumption.
>
> **A3:** Results in Table 3 and Table R1.3 demonstrate that EMLoC is comparable to fine-tuning methods on multiple multi-modal tasks. Those comparison demonstrate that our training-free EMLoC can achieve comparable performance with fine-tuning which needs extra training iterations. Therefore, EMLoC is a promising adaptation method with better efficiency.
>
> Our EMLoC is an efficient adaptation method that makes long-context learning practical even on consumer GPUs, with minimal inference cost. We further compare the time and memory consumption of various compression methods in Table R2 (Reviewer 6UND), where our method shows clear advantages in both inference efficiency and accuracy. For additional analysis, please refer to A2 of Reviewer 6UND and A2 of Reviewer Niy5.
>
> **Q4:** How did you train the model parameters for the context compression? Is it just fine-tuning the model for ImageNet?
>
> **A4:** EMLoC is a training-free adaptation method that adaptively searches for the optimal pruning strategy under a JS divergence constraint, without fine-tuning the pretrained MLLM.  For each task, a distinct pruned KV cache is generated and loaded into memory during inference, **eliminating the need for model retraining or redeployment**.
>
> **Q5:** Figure 1 seems less intuitive. Rather than using demonstration examples on the X-axis, why not use context length?
>
> **A5:** Thanks for the suggestion! We have revised Figure 1 by replacing the X-axis with context length, making it more intuitive in illustrating our advantages in both performance and efficiency.
>
> **Q6:** The details of LoRA and full fine-tuning should be more elaborated in the Appendix than in the current manuscript.
>
> **A6:** In LoRA adaptation, we apply LoRA adapters to all linear modules of the LLM, including qkv_proj, out_proj, up_proj, and down_proj, while keeping the vision encoder and multi-modal projector frozen.  The rank and alpha are set to 16 and 32, respectively. In full fine-tuning, only the LLM is fine-tuned with  DeepSpeed ZeRO-3, leaving other parameters frozen. Other unspecified settings follow the default configurations in LLaMAFactory.

---

### Official Review · Reviewer_6UND · 2025-03-13

**Overall Recommendation:** 3

**Summary:**

Following the improvements brought by in-context examples in multi-modal LLMs, the context length compression has becomed a hot topic to make the technique more scalable.
This paper tackles the challenge by introducing layer-wise adaptive pruning, it also provides theoretical justification by doing this layer by layer through Jenseon-Shannon divergence constraint. The proposed method - EMLoC shows onpar or better performance against naive long-context approaches on vairous vision-language benchmarks.

**Claims And Evidence:**

1. I think one of the underlying assumption of chunk-wise compression is each chunk contains several examples as shown in experiment details. How does the author makes sure each example has the same length?

**Essential References Not Discussed:**

The key contribution is context compression technique while the author didn't discuss its relation/advantage over KV-cache algorithms such as PyramidInfer[1] and FastGen[2]

[1] PyramidInfer: Pyramid KV Cache Compression for High-throughput LLM Inference
[2] Model Tells You What to Discard: Adaptive KV Cache Compression for LLMs

**Experimental Designs Or Analyses:**

I have examined the experimental set up for various number of examples used in vision-language benchmarks.

**Methods And Evaluation Criteria:**

I am not very familiar with vision-language benchmarks but the experiment setups makes sense to me.

**Other Comments Or Suggestions:**

N/A.

**Other Strengths And Weaknesses:**

The method seems cannot be applied together with model parallelization techniques because of the way layer pruning is implemented.

**Questions For Authors:**

1. Can you provide examples, and chunks examples to help me understand how chunk-wise segmentation is possible in vision-language tasks?

2. How about the performance/efficiency improvements with other KV cache techniques proposed for transformers?

**Relation To Broader Scientific Literature:**

It is related to KV cache optimization in the LLM research.

**Theoretical Claims:**

I scan the JS constraint proof and finds it is correct.

---

> ### Author Rebuttal · Authors · 2025-03-31
>
> **Q1:** Each chunk contains several examples as shown in experiment details. How does the author ensure each example has the same length?
>
> **A1:** Thank you for your comment. In ImageNet100, 200 multi-modal examples are evenly divided into 10 chunks. Each image (224×224) is encoded into approximately 64 tokens (may vary slightly due to dynamic aspect ratios), and the corresponding question-answer pair adds around 20 tokens, resulting in about 80 tokens per example. Each chunk contains 20 examples (roughly 1.6k tokens). The system prompt appears only at the start of the first chunk. Below is an example structure:
>
> ```python
> # Start of 1st chunk
> <|im_start|> system\n You are a helpful assistant.<|im_end|>
> ## sample 1
> <|im_start|> user\n <|vision_start|> <Image1.jpg> <|vision_end|> What category does the image belong to? <|im_end|>
> <|im_start|> assistant\n <class 1>. <|im_end|>   ...
> # Start of 2nd chunk
> ## sample 21
> <|im_start|> user\n <|vision_start|> <Image21.jpg> <|vision_end|> What category does the image belong to? <|im_end|>
> <|im_start|> assistant\n <class 11>. <|im_end|> ...
> ```
>
> For other image benchmarks, each image is encoded into 256 tokens(448×448 resolution). Each chunk has 4 examples, resulting in a chunk size of 1.1k–1.6k. For the YouCook2 video benchmark , each video with 8 frames is encoded into 1024 tokens, with 4 videos per chunk, yielding a 4.7k chunk size. If sample lengths vary significantly, we use a greedy algorithm to progressively fill each chunk up to a maximum size.
>
> **Q2:** The key contribution is the context compression technique while the author didn't discuss its relation/advantage over KV-cache algorithms such as PyramidInfer and FastGen. How about the performance/efficiency improvements with other KV cache methods?
>
> **Table R2: Comparison with other context compression methods on ImageNet100**
>
> | Method       | Retention Ratio | Adapt Time | Adapt Memory | Infer Time | Infer Memory | Acc      |
> | ------------ | --------------- | ---------- | ------------ | ---------- | ------------ | -------- |
> | MLoC         | 100%            | 28s        | 62G          | 31m        | 19G          | 62.6     |
> | PyramidKV    | 22.4%           | 54s        | 34G          | 19m        | 17G          | 49.3     |
> | FastGen      | 36.0%           | 45s        | 38G          | 37m        | 21G          | 49.3     |
> | PyramidInfer | 24.6%           | 41s        | 42G          | 21m        | 17G          | 55.6     |
> | **EMLoC**    | **22.4%**       | 144s       | 38G          | **18m**    | **17G**      | **63.7** |
> | **EMLoC***   | 27.6%           | 85s        | **24G**      | 19m        | **17G**      | 60.9     |
>
> **A2:** Thanks for the comments. In Section 4.3 and Table 4 of the original paper, we compared our adaptive EMLoC with two static KV-cache algorithms. Table R2 extends this comparison (Table 4) by including PyramidInfer and FastGen. Most KV-cache methods focus on uni-modal text compression, but fail to maintain original performance with a high compression ratio. **EMLoC retains only 22.4% of tokens while achieving 63.7% accuracy**, outperforming FastGen (49.3% accuracy with 36% tokens) and PyramidInfer (55.6% accuracy with 24.6% tokens). Unlike existing KV-cache methods, EMLoC effectively maintains the full-context performance while significantly reducing the context length, thus improving efficiency.
>
> To optimize the trade-off between adaptation cost and inference performance, we explore **increasing the chunk count (10 → 20)** and a **group-wise strategy** (every two layers share the same retention ratio). This variant, **EMLoC\***, reduces adaptation time from 144s to 85s and memory from 38G to 24G, at the cost of a slight accuracy degradation(63.7 → 60.9) and a higher retention ratio (22.4% → 27.6%). This allows for a flexibility implementation on computation constrained scenarios. The adapation cost is significantly smaller compared with its gains in inference efficiency. More discussion can be seen in the response to Q2 from Reviewer Niy5.
>
> **Q3:** Can this method be applied alongside model parallelization?
>
> **A3:** Yes, the proposed method is compatible with model parallelization. EMLoC only compresses the key-value cache of the context at each layer, and it does not alter the model weights or architecture. Modern model parallel frameworks, using pipeline or tensor parallelism, can distribute the KV cache across devices, allowing for parallelization techniques during EMLoC's adaptation or inference. Varying KV-cache lengths across layers may cause the computational or communication imbalance. To mitigate this, we can manually split the model based on the pruned KV-cache lengths, or automatically share free GPU resources with other models using Multi-Process Service (MPS)[1] or Transparent GPU Sharing (TGS)[2].
>
> [1] NVIDIA MPS: https://docs.nvidia.com/deploy/mps/index.html
> [2] Transparent GPU sharing in container clouds for deep learning workloads. (NSDI 23)

---

### Official Review · Reviewer_Pa96 · 2025-03-14

**Overall Recommendation:** 3

**Summary:**

This paper introduces Efficient Multi-Modal Long Context Learning (EMLoC), a training-free approach that embeds many demonstration examples into large multi-modal inputs, then uses chunk-wise compression and layer-wise adaptive pruning to reduce the resulting key-value cache. By enforcing a Jensen–Shannon divergence threshold at each layer, EMLoC selectively retains important tokens without re-training the underlying model. Experiments on multiple vision-language tasks show that EMLoC preserves or improves performance. Experiments on ImageNet even show the training-free adapted model is comparable to the fine-tuned models.

**Claims And Evidence:**

1. EMLoC outperforms existing long-context models in efficiency and effectiveness. Not compared against other multi-modal in-context learning methods in Table 1.

2. EMLoC generalizes well to different multi-modal tasks. No evaluation on long video understanding or effective retrieval within video context. Test on long video benchmarks to confirm seamless adaptation.

3. EMLoC significantly reduces computational overhead while maintaining high performance. Supported by Line 325-328 experiments (FLOP and inference time reduction).

4. EMLoC outperforms LoRA and achieves performance comparable to full fine-tuning. Only tested on ImageNet100, making the claim too strong. Expand evaluation to more diverse multi-modal benchmarks.

**Essential References Not Discussed:**

N/A

**Experimental Designs Or Analyses:**

1. EMLoC is compared to fully fine-tuned and LoRA-based tuned models. However, it is only tested on ImageNet100. Please expand the experiments and evaluation to make your claim stronger.

2. In terms of training-free long-context multi-modal learning, please compare against LongVA and experiment on some long-video benchmarks.

Zhang, Peiyuan, et al. "Long context transfer from language to vision." arXiv preprint arXiv:2406.16852 (2024).

**Methods And Evaluation Criteria:**

1. Relevance of chunk-wise compression and pruning: The methodology directly addresses the challenge of handling very long multi-modal inputs within limited computational resources. This aligns well with the stated goal of “efficient multi-modal long-context learning.”

2. Choice of benchmarks: While the selected benchmarks do test multi-modal capabilities, they do not fully cover long video tasks or extensive multi-image retrieval scenarios. This partially demonstrates EMLoC’s potential but leaves out broader, real-world applications requiring extended temporal context.

**Other Comments Or Suggestions:**

N/A

**Other Strengths And Weaknesses:**

N/A

**Questions For Authors:**

Please see my suggestions in the above sections related to methods and claims, and experimental designs.

**Relation To Broader Scientific Literature:**

1. Training-free adaptation resonates with in-context learning trends. Whereas concurrent works fine-tune or add adapters for new tasks, EMLoC echoes the literature advocating prompt-only or retrieval-style adaptation, with the unique twist of compressing multi-modal exemplars directly in the model’s KV cache.

2. Bridging multi-modal ICL. Flamingo and other models have shown the possibility of in-context learning for multi-modal tasks but do not specifically address extremely long input contexts. EMLoC advances this line of work by integrating compression/pruning for more scalable “many-shot” demonstration examples.

**Theoretical Claims:**

The mathematical proofs in Sec. 3.3 appear generally sound.

---

> ### Author Rebuttal · Authors · 2025-03-31
>
> **Q1:** Comparison with other multi-modal in-context learning methods in Table 1.
>
> **Table R1.1: Comparison with multi-modal in-context learning methods**
>
> | Method | ImageNet100 | MME-RW   | OK-VQA   |
> | ------ | ----------- | -------- | -------- |
> | MTV    | 32.7        | 27.8     | -        |
> | RICES  | **64.5**    | 40.5     | 58.5     |
> | EMLoC  | 63.7        | **42.2** | **58.7** |
>
> **A1:** Thanks for the suggestion! We have compared EMLoC with two other open-sourced multi-modal in-context learning methods, RICES[1] and MTV[2], on ImageNet100, MME-RW, and OK-VQA in Table R1.1. RICES retrieves the top 1/4 most relevant in-context samples from all samples. MTV extracts the mean activation of in-context examples as task vectors and finds the optimal replacement position of these task vectors. During inference, MTV replaces these task vectors at the optimal position of the test sample, which fails to facilitate these tasks. Our **EMLoC achieves better average performance** across the three benchmarks. It's worth noting that RICES is an online retrieval-augmented method, so it needs to forward the retrieved long context during each inference step. **RICES takes 5 hours inference time and 43G memory cost on ImageNet100, while our EMLoC requires only 18 minutes with 17G memory**, showing clear advantages in efficiency.
>
> [1] Flamingo: a visual language model for few-shot learning (NeurIPS 22)
>
> [2] Multimodal Task Vectors Enable Many-Shot Multimodal In-Context Learning (NeurIPS 24)
>
>
>
> **Q2:** Comparison with LongVA and experiment on long-video benchmarks.
>
> **Table R1.2: Comparison with LongVA and MLoC on VideoMME w/o subtitles with 384 frames**
>
> | Method | Context Length | LLM FLOPs  | LLM Time | Peak Memory | Overall ACC |
> | ------ | -------------- | ---------- | -------- | ----------- | ----------- |
> | LongVA | 55.5k          | 1715.5T    | 22h      | 41G         | 51.8        |
> | MLoC   | 27.9k          | 554.8T     | 7h       | 38G         | **60.3**    |
> | EMLoC  | 2.3k           | **272.0T** | **5h**   | **24G**     | 60.1        |
>
> **A2:** Thank you for the insightful suggestion. As required by the reviewer, we have conducted experiments on the long-video benchmark VideoMME without subtitles, using 384 frames per video. In LongVA, each frame consists of 144 tokens, whereas in Qwen2-VL, 144 tokens represent every two frames through temporal pooling. Compared to our baseline MLoC, EMLoC significantly **reduces computational overhead while maintaining nearly the same accuracy**. Specifically, EMLoC reduces the average context length from 27.9k to just 2.3k tokens, LLM FLOPs from 554.8T to 272.0T, inference time from 7 hours to 5 hours, and peak GPU memory from 38G to 24G, while preserving a consistent accuracy (60.1 vs. 60.3).
>
> To achieve this efficiency, we set $\delta$ =0.04 and configured the retention ratio to \[0.02, 0.1, 0.5, 1.0]. Instead of optimizing the retention ratio for each layer individually (layer-wise), we adopt a **group-wise strategy**, where every 14 layers are treated as a single group and share the same retention ratio. This allows for a more stable and efficient selection process during online inference. Under identical setup(384 frames at the same resolution), both MLoC and EMLoC outperform LongVA while requiring significantly fewer computations. EMLoC also enables the real-time long-video understanding on consumer-grade GPUs such as the NVIDIA 3090, making it a more practical solution for real-world applications.
>
>
>
> **Q3:** Expand evaluation to more diverse multi-modal benchmarks.
>
> **Table R1.3: Comparison with fine-tuning methods on more multi-modal benchmarks**
>
> | Method           | ImageNet100 | MME-RW | OK-VQA | Average  |
> | ---------------- | ----------- | ------ | ------ | -------- |
> | LoRA             | 61.1        | 42.1   | 60.9   | 54.7     |
> | Full Fine-tuning | 64.7        | 42.7   | 49.7   | 52.4     |
> | EMLoC            | 63.7        | 42.2   | 58.7   | **54.9** |
>
> **A3:** We appreciate the reviewer’s suggestion for further benchmarking. To this end, we have compared EMLoC with LoRA and full fine-tuning on additional multi-modal benchmarks(MME-RW and OK-VQA) as seen in Table R1.3. Our EMLoC shows comparable performance to LoRA and full fine-tuning in a training-free manner. This flexibility is essential in real-world applications, where fine-tuning may not always be feasible. The optimization steps and hyperparameters are the same as depicted in the Appendix C.1.

---

### Decision · Program_Chairs · 2025-05-01

**Decision:**

Accept (poster)

**Comment:**

This paper introduces EMLoC, a training-free method for efficient long-context multimodal learning via adaptive KV cache compression. While initial reviews raised concerns about computational efficiency (Niy5) and comparisons (6UND), the authors' rebuttal convincingly addressed these. The reviewers acknowledged the improvements, with iyMg upgrading to weak accept. The work's ​​clear technical contribution​​ and ​​strong empirical validation​​ justify acceptance. The AC recommends acceptance of the paper, and suggest the authors to further answer reviewer Niy5's question on explaining in more detail why those hyperparameters perform well across different datasets. Congratulations.